# Effects of Silirum^®^-Based Vaccination Programs on Map Fecal Shedding and Serological Response in Seven French Dairy Herds

**DOI:** 10.3390/ani13091569

**Published:** 2023-05-08

**Authors:** Fabien Corbiere, Dorra Guellouz, Christian Tasca, Laurent Foures, Emma Dubaux, Gilles Foucras

**Affiliations:** 1INRAE, ENVT, UMR 1225 IHAP, 31076 Toulouse, Francegilles.foucras@envt.fr (G.F.); 2GDS Meuse, 55108 Verdun, France

**Keywords:** paratuberculosis, vaccination, serum ELISA, fecal qPCR, longitudinal study, controlled field study

## Abstract

**Simple Summary:**

Paratuberculosis is an economically important disease in ruminants, and control in affected herds/flocks primarily relies on test and cull strategies and good hygiene and management practices. Vaccination against the causative agent *Mycobacterium avium* subsp. *paratuberculosis* is also used as a tool in some paratuberculosis control programs. The effects of the whole-cell heat-killed Silirum^®^ vaccine on bacterial shedding in the feces and serological response were studied in a controlled field study involving seven French dairy herds, where calves were vaccinated at various ages. A statistically significant reduction of the probability of fecal shedding was observed for cows vaccinated before 4 to 5 months of age compared to non-vaccinated controls. A strong effect of age at vaccination on the serological status investigated in adulthood was also demonstrated, which was also associated with a difference in the protective effect of vaccination.

**Abstract:**

(1) Background: paratuberculosis is an important disease in ruminants, causing worldwide economic losses to the livestock industry. Although vaccination is known not to prevent transmission of the causative agent *Mycobacterium avium* subsp. *paratuberculosis* (Map), it is considered an effective tool for paratuberculosis in infected herds. The objectives of this controlled field study were to evaluate the effects of the whole-cell heat-killed Silirum^®^ vaccine on Map fecal shedding and serological status in dairy herds infected with paratuberculosis. (2) Methods: The serological status (ELISA) and fecal shedding (qPCR) of 358 vaccinated cows were assessed over 3 years in 7 infected dairy herds in the Meuse department, France. Within each herd, cows from the last non-vaccinated birth cohort (*n* = 265) were used as controls. The probability and level of Map fecal shedding and the serological status were modeled using multivariable mixed general linear regression models. (3) Results: Overall, 34.7% of cows tested positive at least once on fecal qPCR, with significant differences between herds, but high shedding levels were observed in only 5.5% of cows. Compared to non-vaccinated seronegative cows, a statistically significant reduction in the probability of Map shedding was found only in cows vaccinated before 4 to 5 months of age that tested negative for Map antibodies throughout the study period (odds ratio = 0.5, 95% confidence interval: 0.3–0.9, *p =* 0.008), but no significant effect of vaccination on the amount of Map shedding could be evidenced. Finally, the younger the cows were when vaccinated, the less they tested positive on the serum ELISA. (4) Conclusions: a beneficial effect of vaccination on Map fecal shedding may exist in cows vaccinated before 4 to 5 months of age. The variability of the serum ELISA response in vaccinated cows remains to be investigated.

## 1. Introduction

Paratuberculosis (PTB), also known as Johne’s disease, is a severe slow-developing, and incurable proliferative enteritis of cattle, sheep, goats, and other domestic and wild ruminants caused by the intracellular bacterium *Mycobacterium avium* subsp. *paratuberculosis* (Map) [1]. Animals with clinical signs of the disease are generally euthanized or culled from the herd, and sub-clinically infected animals also tend to have reduced milk production [2], a longer calving-to-conception interval [3], and an increased likelihood of culling, leading to large economic losses [4,5]. The control of paratuberculosis is largely hampered by the lack of sensitivity of available diagnostic tests, especially for the detection of sub-clinical infections (i.e., clinically healthy animals) [6], which results in animals shedding Map in their environment without being identified. Moreover, Map remains viable for weeks to months in the environment, facilitating the exposure and contamination of susceptible animals [7,8,9]. Numerous programs have been developed in many countries to control PTB in domestic animals, and to reduce the within and between herd/flock transmission of Map [10]. At the herd level, control strategies aim to eliminate infected animals from the herd (particularly those affected and infectious—e.g., test-and-cull strategy) to break down the transmission routes of the disease and to reduce the risk of infection, particularly for young animals, by applying strict hygiene and management measures. In some cases, these strategies are successful in reducing the number of subclinically and clinically affected animals [11,12,13]. However, successful disease control requires the commitment of owners and herd managers over many years. When effective changes are lacking or reduced, clinical cases of PTB continue to occur years and even decades after a control strategy has been put in place [14]. Moreover, the compliance of farmers with recommendations on the control of paratuberculosis can be low, especially when clinical cases have disappeared [15]. Amongst other control strategies, considerable effort has been made to identify candidate genes associated with reduced susceptibility or increased resilience to Map infection, primarily in dairy cattle breeds [16]. Although results are not consistent among studies, these control strategies appear promising [17,18,19,20]. It may, however, be years or decades before the benefits of such control strategies can be evaluated, and research studies have only focused on a few breeds.

Vaccination against paratuberculosis was first described as early as 1926 [21], at which time live vaccines were used. Since then, numerous studies have been performed to evaluate the effects of vaccination against Map in cattle, sheep, and goats. It has been demonstrated to be, on average, effective in reducing the clinical incidence of PTB, delaying the onset of the disease and intestinal lesions and reducing fecal shedding of Map [22]. However, vaccination is not widely used due to its possible interference with intradermal testing for bovine tuberculosis [23,24,25], and is currently allowed in only a few countries around the world [10]. Vaccination is also known to induce a long-lasting antibody response that may interfere with the use of serological tests for the detection of PTB-infected animals [26,27]. In France, after the production of the Neoparasec^®^ vaccine was discontinued in 2001 [28], vaccination against paratuberculosis was not possible, until the Silirum^®^ vaccine (CZ Veterinaria, S.A., Porriño, Spain) was approved in 2014. Its use is, however, only allowed in herds deemed to be free of bovine tuberculosis and where the presence of Map has been confirmed and is subject to authorization by the veterinary authorities. Silirum^®^ is a whole-cell heat-killed Map (strain 316F) in an oil adjuvant. Field studies reporting its effect on the epidemiological dynamic of PTB are scarce and almost all come from the Basque Country, Spain, where it has been widely applied for almost two decades. A significant reduction in the number of culled cows showing lesions associated with PTB and with diffuse or severe PTB lesions was found in vaccinated cows compared to non-vaccinated controls in a single herd where cattle were vaccinated regardless of their age or clinical status [28]. Similarly, a significant attenuation of pre-existing infection in naturally PTB-infected cows vaccinated as adults has been suggested by Alonso-Hearn et al. [29]. Juste et al. (2009) [30] reported a decline in the number of Map shedders and the total amount of Map shed 2 to 4 years after all cattle were vaccinated in 6 dairy herds. This reduction in Map shedding was also demonstrated in animals vaccinated when less than 6 months old or at adult age, with a significant difference depending on the age of vaccination [31]. Based on the analysis of a large cohort from several vaccinated and non-vaccinated herds, the same research group also suggested a specific and non-specific effect of vaccination on cattle lifespan [32]. The Gudair^®^ vaccine (CZ Veterinaria, S.A., Porriño, Spain), related to the Silirum^®^ vaccine and approved for use in sheep and goats, has been intensively used in Australia for almost two decades, and is effective in reducing the incidence of clinical cases, the prevalence of lesion attributable to ovine paratuberculosis, and the prevalence of Map shedding in affected flocks [33,34].

At the initiative of the Groupement de Défense Sanitaire de la Meuse (GDS Meuse), a voluntary vaccination program was launched in 2015 in the Meuse administrative department, France, as a potential aid to control losses associated with paratuberculosis in affected herds already engaged in a control plan. The aim of the present study was, therefore, to evaluate the effects of Silirum^®^-based vaccination programs on Map fecal shedding and serological response in a subset of these herds. By comparing a cohort of vaccinated and unvaccinated (control) cows reared under the same management conditions within each herd, we sought to evaluate vaccination as a management tool in paratuberculosis control programs.

## 2. Materials and Methods

### 2.1. Herd and Cows Selection

Seven dairy herds were selected on a voluntary basis. Herds were infected with Map and involved in the paratuberculosis control plan managed by GDS de la Meuse for between 6 to 18 years at the beginning of the current study. They were amongst the first to join the paratuberculosis vaccination plan launched by GDS de la Meuse after Silirum^®^ was approved in France in 2014. The paratuberculosis control plan relies on hygiene and management measures aiming at reducing calf exposure to Map, including calving in a dedicated paddock, the immediate removal of calves from their dam after birth (before suckling), feeding with colostrum from a dam that tested negative for Map antibodies, and subsequently, with milk replacer or pasteurized waste milk, raising in separate housing with dedicated equipment, and, after weaning, grazing on dedicated pastures with no contact with adult cattle or adult manure. Before vaccination, test-and-cull measures based on the serological testing of all cows over 24 months of age every six months were also implemented. After the introduction of vaccination within each herd, test-and-cull measures were maintained only for non-vaccinated cows. On some occasions, fecal qPCR was performed to evaluate the shedding status of non-vaccinated cows that tested positive for Map antibodies or on vaccinated cows with clinical suspicion of paratuberculosis as a guide for removal from the herd. 

All herds were free from tuberculosis, Prim’Holstein herds with free-stall housing for lactating cows, ranging in average size from 81 to 302 cattle over 24 months old and for which vaccination against paratuberculosis had been authorized by the French veterinary services.

In the selected herds, vaccination against paratuberculosis was initiated between April 2015 and April 2016, and was performed only in young animals. For logistic and convenience reasons, at the first vaccination date within each herd, all calves older than 1 month and younger than 12 months were vaccinated. Subsequently, all calves older than 1 month were vaccinated when their number reached a multiple of 5 (i.e., the minimum volume of the Silirum^®^ vaccine bottle (CZ Veterinaria, S.A., Porriño, Spain)). This vaccination schedule explains the variation in age at vaccination within each herd. Each calf received a single dose of 1 mL of Silirum^®^, administrated subcutaneously by an accredited veterinarian. Animal id and vaccination date were recorded on a database managed by the GDS de la Meuse.

Within each herd, a sample of cows was selected for the present study. The selected cows had to be born on the farm and be in their second lactation in the first year of the study (i.e., between May 2018 and 2019). Because the current study was initiated in April 2018, 2–3 years after the start of vaccination depending on the farm, some cows meeting these criteria had not been vaccinated (last birth cohort of unvaccinated cows) and were included as intra-herd controls. Overall, in this setting, 358 vaccinated and 265 non-vaccinated cows were included. A one-sided power calculation, assuming that 25% of unvaccinated cows would shed Map in their feces, indicated that these numbers allowed for a one-third reduction in shedding frequency in vaccinated cows to be evidenced, with a statistical power 1 − β = 80% and a type I error α = 5%.

All farmers participated voluntarily and gave written consent that the test results would not be shared with them until the end of the study to avoid bias due to cows being removed from the herd based on a positive result. 

Information related to cows (birth herd, date of birth, and calving dates) was extracted from the French Base de Données National d’Identification (BDNI).

### 2.2. Sample Collection and Handling

Farms were visited on 5 occasions between May 2018 and January 2021 at 6-month intervals, except for the last sampling date, which was delayed due to the COVID-19 lockdown in France. However, because not all cows had entered their second lactation by the first two sampling dates (i.e., May and October 2018), and considering the high renewal rate in dairy herds, the total number of samples collected per cow was typically between 1 and 3 (see Section 3). A handful of feces was sampled from the rectum of selected animals using single-use gloves without lubricant and was placed in an individually identified sterile plastic bag for transportation. In parallel, a five-milliliter blood sample was also collected from the caudal vein in vacuum tubes without anticoagulant (Vacutainer^®^ System). Feces and serum samples were frozen at −20 °C before analysis, which was carried out within two months from sampling. 

Blood and feces samples were collected by accredited veterinarians as part of the routine paratuberculosis control program managed by GDS de La Meuse. All animal owners gave written consent for samples collected from their animals as part of the paratuberculosis control plan managed by GDS de la Meuse to be used in this study. In this context, ethical approval for animal procedures was deemed not necessary by the Ecole Nationale Vétérinaire de Toulouse ethical committee.

### 2.3. Laboratory Testing

#### 2.3.1. Serological Tests

A commercial ELISA test (ID Screen Paratuberculosis Indirect^®^, IDVet, Montpellier, France) was applied to serum samples using an overnight incubation protocol following the manufacturer’s instructions. All samples were tested in duplicate and the average sample-to-positive (S/P) ratios were used as the final result. Duplicates showing discordant results were systematically retested. Negative and positive controls provided by the manufacturers were included on each ELISA plate, and the manufacturer’s guidelines were strictly followed for the interpretation of S/P ratio results: serum samples with S/P values < 60%, between 60 and 70%, and ≥70% were considered negative, doubtful, and positive for Map antibodies, respectively. 

#### 2.3.2. Fecal Real-Time PCR

First, fecal samples underwent a concentration procedure using the ADIAFILTER system (BioX, Rochefort, Belgium) following the manufacturer’s instructions. Ten grams of feces was rehydrated for 1 h in 70 mL of bi-distilled sterile water. The top 10 mL of the supernatant was then filtered and centrifuged using the ADIAFILTER^®^ disposal. Pellets were then resuspended in 500 µL of bi-distilled water and mixed with 300 mg of 100 µm silica beads (PRECELLYS LYSING KITS VK01, Bertin Technologies, Montigny-le-Bretonneux, France) for 30 s at 6800 rpm (i.e., approximately 3200 g) three times in a bead beater (Precellys 24^®^, Bertin Technologies, Montigny-le-Bretonneux, France). Magnetic bead-based DNA extraction was performed on a Kingfisher Flex^®^ magnetic particle processor (Thermo Fisher Scientific, Courtaboeuf, France), following the NucleoMag 96 tissue protocol (Macherey-Nagel, Hoerdt, France), with the addition of an extraction control (ADIAVET™ PARATB REAL TIME, BioX, Rochefort, Belgium) in each plate well. Samples were subjected to qPCR (ADIAVET™ PARATB REAL TIME, BioX, Rochefort, Belgium), following the manufacturer’s instructions. Each sample was also tested for amplification of the internal control. Bi-distilled water and synthetic IS900 DNA provided in the amplification kit were used as negative and positive controls, respectively. In addition, fecal samples from a cow at the clinical stage of paratuberculosis with extensive multibacillary intestinal lesions and from a clinically healthy vaccinated but persistently shedding ewe with only focal paucibacillary intestinal lesions were systematically included in each test plate, following an identical preparation process to that of tested samples. Fifty amplification cycles were performed on a LightCycler 96 (Roche Life Science, Meylan, France), and fluorescent signals were recorded in two channels, with FAM detecting IS900 and VIC detecting the extraction control. Due to the overlapping spectra of the two dyes, a color compensation step was applied. Raw fluorescence data were obtained from the LightCycler 96 and modeled using the qpcR package [35] in R software (version 4.2.2) [36]. Cycle thresholds were determined using the second derivative maximum (CpD2). 

All DNA amplifications were ran in duplicate in separate plates, and the average cycle count (Ct values) was used as the final result. Duplicates yielding discordant results (i.e., absolute difference in Ct values > 2.5) were further retested. 

Samples that reached fluorescence with a Ct value below 40 were considered positive. A careful examination of late fluorescence curves indicated that they were associated with low but unambiguously positive results up to 40 Ct, while non-specific amplification results could not be ruled out beyond this threshold. 

An external quantification curve was constructed using triplicate end-point dilutions of a purified Map culture of the reference K10 strain ATCC 19698 from 10^8^ to 10 copies of the IS900 gene per qPCR plate well, using the DNA amplification protocol, as described above. This quantification curve was used to approximate the amount of Map in each qPCR-positive fecal sample.

### 2.4. Data Analysis

#### 2.4.1. Data Description and Univariable Analysis

Frequencies and detection rates were compared using the Fisher exact test. Quantitative variables were compared using the Student *t*-test or the Wilcoxon–Mann–Whitney test for ranks when the distributions were strongly skewed or when the sample size was small. A Benjamini–Hochberg correction for multiple comparisons was applied when needed. 

#### 2.4.2. Multivariable Analysis

Logistic mixed-effect models were fitted to data available from all cows to model the effect of vaccination on the probability of Map shedding, indicated by a positive fecal qPCR result. The subset of qPCR-positive fecal samples was used to model the effect of vaccination on Map shedding levels, expressed as log10 estimated number of Map per gram of feces (log10 Map·g^−1^). Because the distribution of the response variable was strongly skewed to the right, a gamma error distribution with a log link was used. Fixed effects included age at vaccination (AGEVACC), age at sampling (AGESAMP), days in milk within the sampling lactation (DIM), and the serological status at the time of sampling (SEROSTAT). The effect of quantitative variables (i.e., AGEVACC, AGESAMP, DIM) was modeled using natural B-splines to ensure smoothness in the relationship between the quantitative variables and the outcome or by creating dummy variables with different cutoff values when appropriate. To account for the clustered structure of the data (several sampling points for each cow, cows from different herds), cow and farm random effects were included. 

In both models, the statistical unit was the fecal sample Y*_ijk_* from cow *i* at sampling point *j* within the *k* herd.

A final logistic mixed-effect model was fitted to the subset of vaccinated cows to model the effect of age at vaccination and potential confounders, including AGEVAC, DIM, AGESAMP, and fecal qPCR status, on the probability of yielding a positive serum ELISA result at the first sampling point.

When needed, separate analyses were performed according to whether doubtful ELISA results were handled as positive or negative. 

Data description and analysis were performed using the R software (version 4.2.2) [36]. Generalized linear mixed-effect models were fitted using the glmer function from the “lme4” package [37] and the glmmTMB function [38]. The marginal effects of continuous variables modeled as splines were computed using the “effects” package [39]. Statistical significance was defined as a *p*-value less than 0.05, and the results meeting this criterion were considered significant. 

## 3. Results

### 3.1. Data Description

#### 3.1.1. Herds and Sampled Cows

During the study period, 358 vaccinated and 265 non-vaccinated control cows were sampled, with unequal distribution across herds (Table 1). On average, this represented 48.4% of cattle older than 24 months, present in the 7 herds (between 38.0% and 58.6% depending on the herd). A majority of cows (*n* = 365, 58.6%) were sampled twice, and the proportion of cows sampled only once was significantly higher (*p =* 0.018) amongst the vaccinated cows (31.3%) than the non-vaccinated ones (22.7%) (Table 2). 

The median age at first sampling was 48.0 months (first quartile Q1 = 43.8 months and third quartile Q3 = 53.0 months), and, overall, 254 out of the 1163 samples (21.8%) occurred in cows older than 5 years. On average, vaccinated cows were 7.5 months younger that non-vaccinated controls at the different sampling dates (*p* < 10^−6^). More details on age at sampling and time since vaccination are provided in Appendix A.

The selected cows had received a single dose of Silirum^®^ vaccine at a median age of 4.3 months (Q1 = 2.8, Q3 = 6.9). Significant differences were evidenced across herds (*p* < 2 × 10^−5^), with cows in Herd A vaccinated at a median age of 6.1 months (Q1 = 4.4, Q3 = 9.2), and those in Herd D at a median age of 3.1 months (Q1 = 2.1, Q3 = 4.3).

#### 3.1.2. Serological Results

Overall, 13.0% (*n* = 151) of serum samples were positive for Map antibodies, and 1.7% (*n* = 20) were doubtful. The proportion of positive or doubtful results was significantly higher among vaccinated animals (23.7%) than among non-vaccinated animals (3.8%) (*p* < 10^−6^).

In the 246 vaccinated cows sampled at least twice, 167 (67.9%) remained negative for Map antibodies, 35 (14.22%) were always positive, 25 (10.2%) changed from positive to negative, and 19 (7.7%) inversely changed from negative to positive. For these 19 cows, seroconversion was detected 38.6 +/− 4.3 months after vaccination (minimum: 32.8 months; maximum: 48.0 months). On average, in the 60 cows tested at least twice and that yielded a positive ELISA result on their first sampling date, a significant decline of the S/P value was observed between their first and last test (median difference in S/P values: −23.8, Q1 = −70.0, Q3 = −4.6, *p =* 0.001). This decrease was observed in 39 out of the 60 cows, while for 7 other cows an increase in the S/P values was observed, which was greater than 50%. Conversely, among the 186 cows tested at least twice and that yielded a negative ELISA result on their first sampling date, a slight but significant increase in the S/P value was observed, on average, between their first and last test (median difference in S/P values: 2.2, Q1 = −6.8, Q3 = 15.9, *p =* 0.004). An increase in S/P values was observed in almost half of these cows (*n* = 88, 47.3%), and was greater than 50% in 60 of them, while a decrease higher than 50% occurred in 55 other cows. 

In the 205 non-vaccinated control cows sampled at least twice, only 10 (4.9%) cows seroconverted and 1 (0.5%) changed from positive to negative; all other cows remained negative on all sampling occasions. On average, the variation in the S/P values between the first and last serological test was significantly positive (median difference in S/P values: 3.8, Q1 = −4.6, Q3 = 16.9, *p =* 10^−4^).

The proportion of initially negative cows that mounted an antibody response towards Map that could be detected by ELISA did not significantly differ between vaccinated and non-vaccinated cows (*p =* 0.122).

Seropositive cows were observed in all herds. The proportion of vaccinated cows that tested positive or doubtful for Map antibodies at least once differed significantly between herds, from 13.3% (Herd D) to 68.7% (Herd G). These proportions ranged from 0% (Herd C) to 14.3% (Herd E) for non-vaccinated cows, but the overall small number of ELISA-positive cows (*n* = 20) precluded any formal statistical comparisons between herds. 

#### 3.1.3. Fecal qPCR Results

The interplate coefficient of variation (cv) of Ct values for the positive control of the qPCR kit and the external positive fecal controls (i.e., feces from the clinically affected cow and the clinically healthy shedding ewe) were 3.2%, 2.7%, and 3.1%, respectively.

Of the 1160 fecal samples tested, 276 (23.8%) were positive by fecal qPCR. However, only 36 of them (12.9%), originating from 34 cows, were considered highly positive (i.e., corresponding to an estimated Map concentration ≥ 1000·g^−1^ of feces) (Table 3, Figure 1). Six samples had an estimated Map concentration higher than the clinically affected cow control sample (i.e., 174,000 Map·g^−1^ of feces), and 26 had an estimated Map concentration lower than the clinically healthy shedding ewe control sample (i.e., 8 Map·g^−1^ of feces). Overall, 216 out of 623 cows (34.7%) (38.3% of vaccinated cows (*n* = 137) and 29.8% of non-vaccinated ones (*n* = 79), *p =* 0.03) tested positive at least once during the study period, but this overall result does not account for the unbalanced distribution of cows between herds. 

Out of the 246 vaccinated cows that were sampled at least twice, 146 (59.3%) consistently tested negative for fecal qPCR and 33 (13.4%) consistently tested positive, while the remaining 82 cows (33.3%) had varying results across the different sampling points, either testing positive or negative. In the 205 non-vaccinated control cows sampled at least twice, 141 (68.8%) were consistently negative and 21 (10.2%) were consistently positive.

Cows that tested positive for qPCR were found in all herds, but in significantly different proportions, from 13.4% (11 out of 82, Herd B) up to 56.2% (27 out of 48, Herd G) and 71.2% (104 out of 146, Herd F). High-shedding cows (i.e., estimated Map concentration ≥ 1000·g^−1^ of feces) were found in all herds except Herd C. The detailed distributions of fecal samples and cows according to herd, vaccination status, and estimated Map concentration are provided in Appendix A, respectively.

### 3.2. Effect of Silirum^®^ Vaccination on the Probability of Fecal Map Shedding

None of the logistic mixed regression models that were fitted indicated a significant effect on the odds of Map shedding of either days in milk within the lactation (*p =* 0.56) or age at sampling (*p =* 0.22). In all instances, the herd random effect was highly significant (*p* < 10^−6^), indicating a strong herd effect on the probability of Map shedding. 

Overall, the probability of shedding was lower in vaccinated cows compared to non-vaccinated ones, but this result only approached statistical significance (OR = 0.72, 95% confidence interval: 0.50–1.02, *p =* 0.07).

After adjusting for age at vaccination, a significant reduction in the odds of Map shedding was only observed in cows that were vaccinated before 4 months of age (*n* = 169 cows), when compared to non-vaccinated cows (OR = 0.58, 95% CI = 0.42–0.94, *p =* 0.025) (Figure 2A). Distinguishing between cows vaccinated before and after 5 months of age yielded very similar results, but was associated with a slightly lower fit to the data. 

Finally, the best fit to the data was provided by models also accounting for the serological status. Indeed, a significant reduction in the odds of shedding was observed in cows vaccinated before 4 months of age and that tested negative for Map antibodies throughout the study period (*n* = 137) (OR = 0.52, CI = 0.32–0.85, *p =* 0.008), but not for those that were vaccinated after 4 months of age and/or tested positive at least once on ELISA (*p* > 0.5) (Figure 2B). Once again, distinguishing between cows vaccinated before and after 5 months of age led to very similar results. Results also indicated that among the non-vaccinated group, cows that tested positive at least once on serum ELISA (*n* = 20) were at higher risk to test positive with fecal qPCR compared to those that always tested negative (*n* = 245) (OR = 3.55, 95% CI: 1.61–7.84, *p =* 0.002). 

A strong significant herd effect was found (*p* < 10^−6^), with Herds A and B associated with the lowest odd of a positive result and Herds F and G with the highest. 

Finally, these findings were still found when only samples with an estimated concentration of Map > 100·g^−1^ were considered positive (*n* = 72). Doubtful ELISA results did not influence model outputs and only the results where they were handled as positive are reported. 

### 3.3. Effect of Silirum^®^ Vaccination on the Level of Fecal Map Shedding

This analysis was performed on the subset of qPCR-positive fecal samples (*n* = 276). No difference in the estimated amount of Map shedding could be demonstrated between cows that always tested negative on serum ELISA and vaccinated cows, regardless of their age at vaccination or their serological status (*p =* 0.58). Conversely, non-vaccinated cows that tested positive on serum ELISA at least once had significantly higher shedding levels than all other cows (*p* < 10^−4^). It is noteworthy that high shedders (i.e., estimated Map concentration ≥ 1000·g^−1^ of feces) were observed in all groups (Figure 3). Furthermore, we found that the amount of Map shedding was significantly and positively related to the age at which cows were sampled (*p =* 0.006), whatever their vaccination status (Figure 4). Again, a strong significant effect of herd was observed (*p* < 10^−6^), with Herds C and D associated with the lowest Map loads and Herds F and G with the highest. 

### 3.4. Effect of Age at Vaccination on Serological Status of Vaccinated Cows

The proportion of vaccinated cows that tested positive or doubtful for Map antibodies at first sampling remained low (81 out of 358, 22.6%). We, therefore, sought to determine the factors influencing the variability of the serological response in vaccinated animals. The analysis was limited to the serological results obtained from the first sampling of each cow and was adjusted on age at vaccination, age at sampling, days in milk within the lactation, and the amount of Map shed in the contemporary fecal sample.

Taking into account the strong herd effect (*p =* 10^−4^), only age at vaccination had a significant effect (*p =* 0.002), with a higher probability of being positive in Map antibodies for cows that were vaccinated at an older age (Figure 5). As an illustration, a raw description indicated that 34.4% of cows (45 out of 131) vaccinated after 5 months of age tested positive at their first sampling point, whereas only 15.9% (36 out of 227) of cows vaccinated at a younger age did so (*p =* 10^−5^).

## 4. Discussion

By comparing a cohort of vaccinated and unvaccinated (control) cows reared under the same management conditions within each herd, this field study sought to evaluate vaccination as a management tool in paratuberculosis control programs. To the best of our knowledge, reports on the effect of Silirum^®^-based vaccination programs focusing specifically on animals vaccinated when young and including within-herd unvaccinated control individuals are lacking. In many studies, results from bacteriologic culture or fecal qPCR from vaccinated cows were either compared with results for nonvaccinated animals from other farms or with herd prevalence of Map infection before vaccination was implemented [29,30,31,40,41]. The use of other herds as comparison groups introduces potential bias because the confounding effects of herd management, including practices designed to reduce Map transmission within herd, cannot be entirely controlled without randomization, and comparisons of prevalence within a herd before and after vaccination is initiated ignore the confounding effect of time. Our study was based on a particular vaccination program that targeted only calves under 12 months of age, allowing the use of non-vaccinated animals as controls within each herd. The choice of GDS de la Meuse not to vaccinate animals over 12 months of age at the beginning of the voluntary vaccination program was mainly motivated by financial reasons, but also by the wish to be able to follow the dynamics of infection through serological testing of non-vaccinated cows. Additionally, although a reduction in Map shedding has been described in already infected cows vaccinated as adults [29,30], the control plan managed by the GDS de la Meuse focused mainly on reducing the exposure of young animals to Map. Our choice to focus only on cows within their second lactation at the beginning of the follow-up was motivated by a compromise between the occurrence of Map shedding, which we expected to be higher as cows aged [42], and the high culling rates commonly observed in French dairy herds (average renewal rate in French Prim-Holstein herds 34%, mean lactation number per cow 2.3). In addition, at the initiation of our study in 2018, all first lactation cows had already been vaccinated (in four of the seven herds), precluding the possibility of having a control group of non-vaccinated cows among this lactation number. 

Management changes aiming to reduce the transmission of Map were enacted long before the introduction of vaccination in all herds, and these changes have probably reduced the incidence of infection in young animals. In the year before the introduction of vaccination, the serological incidence in non-vaccinated cows was below 3% in 5 of the 7 herds, suggesting that the infection pressure may be low, as also supported by the low frequency of cows that seroconverted during the follow-up in the non-vaccinated control group (*n* = 10, 4.9%) or the vaccinated group (*n* = 19, 7.7%). It was not a primary objective of the study to measure the effect of vaccination alone on the change in the herd prevalence of Map shedding, given that most herd farmers would use vaccination in combination with other disease control actions. This low infection rate may have reduced the possibility of evidencing a strong effect of vaccination, as would be expected in herds with high clinical incidence [31,32,42,43]. This reduction in the infection pressure may have impacted our findings with respect to detecting differences between vaccinates and controls. Similarly, in studies in which all animals except controls are vaccinated [32], an indirect protection of non-vaccinated cows may bias the results towards the null, by reducing the environmental bacterial loads to which they are exposed within the herds. This potential bias can be excluded in our study because adult cows were not vaccinated, and also because the cows used as controls were born a few months before the cohort of calves that comprised the vaccinated group. Conversely, vaccinated cows may not have benefited from herd immunity [44]; with the greater environmental pressure than if all animals in the herd had been vaccinated, the possibility of detecting a protective effect of the vaccine is reduced. 

Our findings that vaccinated cattle had detectable fecal shedding less often compared to controls contribute to the growing body of knowledge on the role of vaccination in achieving paratuberculosis control. However, it is well-established that vaccination does not guarantee the complete prevention of actual infection in animals, and that there exists a possibility of high shedding among vaccinated animals [22,29,30,42]. Our results are consistent with these findings, with some cows having fecal samples with high concentrations of Map regardless of their vaccination status or age at vaccination. It is noteworthy that non-vaccinated cows that tested positive on serum ELISA at least once had, on average, significantly higher amounts of Map in their feces than all other cows. Although no individual information was collected regarding clinical status at the time of sampling, we can speculate that some of these unvaccinated ELISA-positive cows were at or near the clinical disease stage [6,45]. If vaccination was able to reduce or delay the onset of clinical paratuberculosis, as reported by others [22], a direct beneficial effect on the overall amounts of Map excreted within a herd could be expected. We also found a significant effect of aging on the amount of Map shedding by infected cows, including vaccinated ones, which raises the question of the duration of the protection offered by the vaccine. Although this issue has been little studied, it has been pointed out in longitudinal studies in sheep or goats vaccinated with the killed whole-cell Gudair^®^ vaccine that it could last over the entire production life of animals [33]. This question, however, still requires further work in cattle for the Silirium^®^ vaccine. The fact that a few vaccinated cows may shed large amounts of Map in their feces, even among those testing negative for Map antibodies, is an argument for maintaining vaccination and good management practices for a very long period of time, even after clinical cases have disappeared, before a beneficial effect on Map transmission dynamics may be evidenced [46,47]. 

A significant effect of the age at vaccination was observed, with a beneficial effect on the odd of Map shedding demonstrated only in cows vaccinated before 4 to 5 months of age. Our findings are in accordance with other studies [31]; the age at vaccination may play an important role in the mounting of an immune response capable of delaying disease progression at the individual level, or that the exposure to Map during a longer period of time in animals vaccinated at older ages may reduce vaccine efficiency [48,49]. Experimental and field studies in lambs, goats, and calves have demonstrated that vaccination may be effective when administrated in animals before or around 1 month of age [42,50,51,52]. However, a decrease in Map shedding has also been shown in cattle vaccinated as adults and presumably already infected [29,30]. Future field and experimental studies focusing on the protective and/or therapeutic effect of vaccination according to the age at which it is administered would be of interest, as this could have practical implications for defining the best vaccination schedule. 

In all models, a strong herd effect was evidenced, meaning that conclusions drawn on the overall effect of vaccination may not be valid in all herds. Interestingly Herds F and G were associated with both a higher probability of shedding and a higher amount of Map shedding in feces, even in vaccinated cows, suggesting that paratuberculosis control was less efficient in those herds. Common management practices designed to reduce Map transmission within each herd were monitored, and farmers were interviewed regularly to ensure that all the producers implemented them similarly, but departures from the best practices may have happened occasionally or repeatedly, which may explain different trajectories at the herd level [47]. Others factors that may influence animal immunity, such as nutritional factors, worm burdens, lameness, or the lack of general cleanliness, which favors the maintenance and transmission of Map, could also contribute to differences between farms, but were not investigated.

Vaccination with either attenuated or inactivated paratuberculosis vaccines is known to induce a strong and long-lasting humoral response, but results vary widely according to the vaccine used or age at vaccination. Interindividual differences are also often reported. In a longitudinal field study involving calves vaccinated before 35 days of age with Mycopar^®^ killed whole-cell vaccine, antibody response was still detected, as assessed by serum ELISA, up to 7 years after vaccination in some animals and, conversely, about one third of the vaccinated animals did not appear to develop a serological response [27]. In lambs vaccinated before 3 months of age with the killed whole-cell Gudair^®^ vaccine, only 15% to 50% of sheep still tested positive on the serum ELISA 2 to 4 years post-vaccination, with large differences between flocks [26]. In contrast, in another field study of 5 flocks where lambs had been vaccinated at 3–6 months old with the same vaccine, the vast majority (>95%) still tested positive on the serum ELISA five years post-vaccination, with no decline in S/P values with age [53]. Interindividual heterogeneity in both humoral and cellular response has also been reported in goats during the first months after vaccination with Gudair^®^ [49]. Traditionally, this humoral response has not been associated with protection against Map infection, as opposed to cellular response [54,55], but a recent experimental study in sheep pointed out that a strong initial B cell response soon after vaccination is associated with the clearance of infection [56]. In our study, only a small proportion of cows tested positive on the serum ELISA (22.6% at first sampling time), and no significant protective effect of vaccination on the odd of Map shedding was observed for those cows that tested positive on the ELISA during the study period, although this finding could be due to a lack of statistical power in relation with the small sample size. The individual trajectories of post-vaccination humoral responses were highly diverse, with a decrease in the ELISA S/P values observed in some vaccinated animals, while no notable variation was observed in others. This low proportion of seropositive animals could be related to the fact that the humoral response was only evaluated in adulthood, 4 to 6 years after vaccination, and does not allow inference about the earlier history of this response. Some of the positive results observed in the ELISA test may also correspond to animals that have responded to a natural infection. However, their proportion appears to be too high, particularly compared to non-vaccinated cows, for this phenomenon alone to be invoked. Interestingly, we found that the younger the cows were vaccinated, the less they tested positive on the serum ELISA, in agreement with previous experimental studies in lambs and kids [49,57]. This supports the hypothesis that the antibody response induced by the paratuberculosis vaccine in young animals is lower than in older ones due to incomplete development of their immune system [58]. Vaccination also induces a Th1/IL-17 immune response [25,59], which is known to play a major role in the protective response against intracellular mycobacterial pathogens. Our findings, therefore, do not imply that the protective effect of vaccination in young animals is lower, as also supported by the significant protective effect of vaccination on the odds of Map shedding found in cows vaccinated when younger than 4 to 5 months of age. Additional research is needed, including the investigation of potential genetic factors [60], to more accurately evaluate the inter-individual variability in serological responses following vaccination and to explore possible links with the efficacy of the vaccine.

## 5. Conclusions

Altogether, our findings contribute to the growing body of knowledge on the role of vaccination in achieving paratuberculosis control, even in herds where good management and hygiene practices are already enacted. The fact that the beneficial effect of vaccination was only demonstrated in cows vaccinated before 4–5 months of age also provides practical information for the management of vaccination schedules in infected herds. Our results also reinforce the idea that even if beneficial in a few years, long-term vaccination is necessary to reduce the risk of new infections.

Finally, non-specific effects of vaccination against paratuberculosis on lifespan have been suggested in cattle and goats [32,49], which could also add to the beneficial effect of vaccination in infected herds. Although still controversial [61], a protective effect against general mortality unrelated to tuberculosis has similarly been suggested by several studies in children vaccinated with BCG or against measles [62,63,64,65]. Other specific and non-specific beneficial effect of vaccination, including increased milk yields and carcass weights, were reported [29,30,32]. The primary objectives of the present study did not include the evaluation of the effect of vaccination on cattle lifespan and production performances, but data are currently being collected in all herds involved in the vaccination plan managed by the GDS de la Meuse to investigate these topics.

## Figures and Tables

**Figure 1 animals-13-01569-f001:**
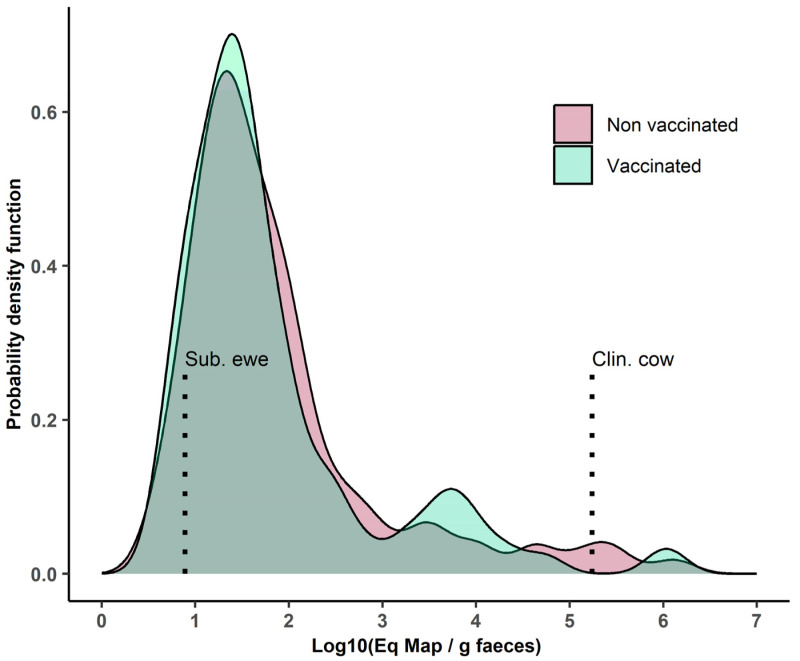
Distribution of the estimated amount of Map shed in the feces of qPCR-positive samples (in log10 equivalent Map·g^−1^), given the vaccination status. To aid interpretation, non-vaccinated cows are shown in red and vaccinated cows are in green. Vertical dotted lines: values for a clinically affected cow (Clin. cow) and a vaccinated clinically healthy but persistently shedding ewe (Sub. ewe).

**Figure 2 animals-13-01569-f002:**
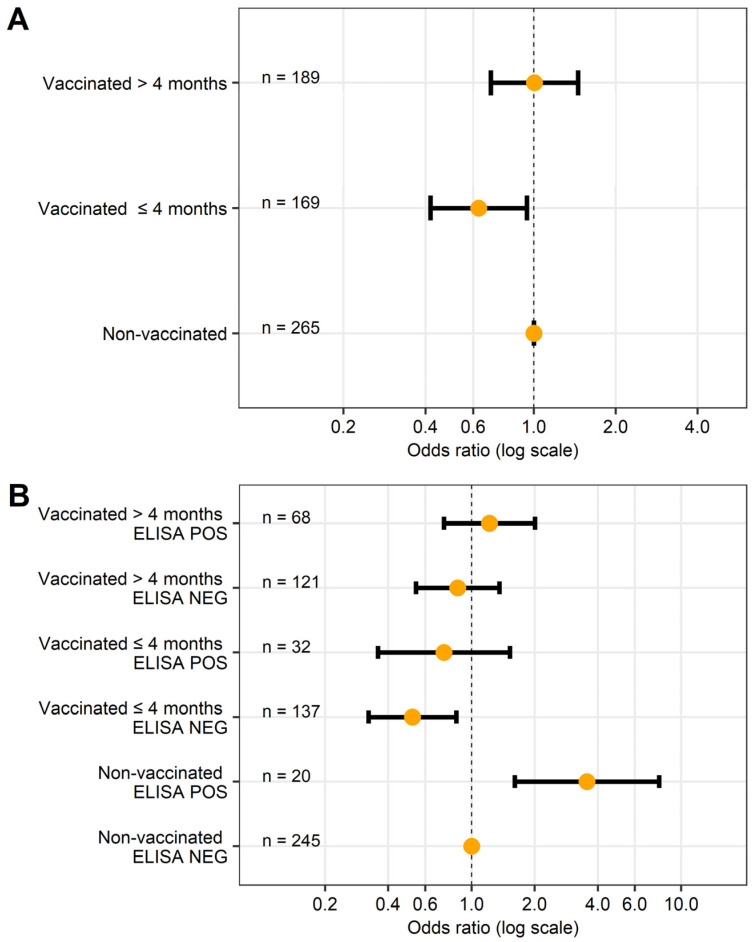
Graphical representation of the effect of vaccination on the odds of a positive result on fecal qPCR. Results are expressed as odds ratio (OR) (orange points) and 95% confidence interval (horizontal black lines). In both panels, the lowest line represents the reference group (OR = 1). The number of cows in each group is indicated within each line. (**A**) Results from the final logistic mixed regression model accounting for vaccination status and age at vaccination. (**B**) Results from final logistic mixed regression model accounting for vaccination status, age at vaccination, and serological status.

**Figure 3 animals-13-01569-f003:**
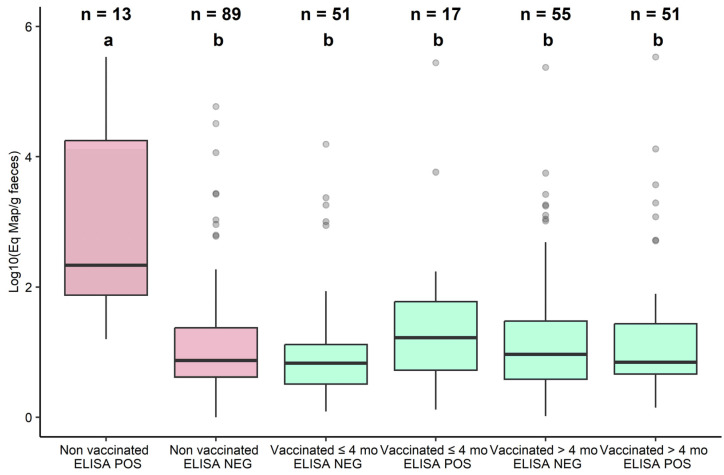
Distribution of the estimated amount of Map shedding in the feces of qPCR-positive samples, given the vaccination status, age at vaccination, and serological status of cows. To aid interpretation, non-vaccinated cows are shown in red and vaccinated cows are in green. The number of fecal samples in each group is reported at the top. Groups not sharing the same letter are statistically different.

**Figure 4 animals-13-01569-f004:**
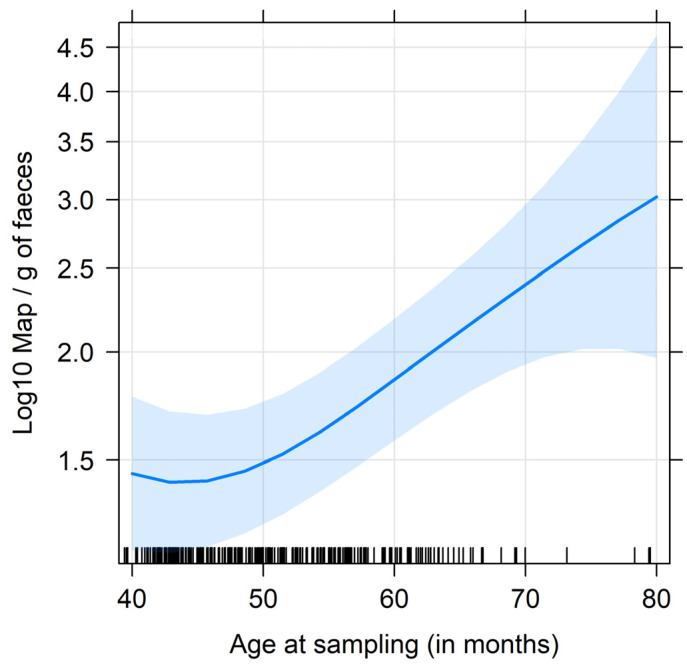
Estimated effect of age at sampling on the amount of Map shedding in the feces. The point estimate is represented by the solid blue line, and its 95% confidence interval is shown by the blue area surrounding the line. The black marks on the bottom represent the observations, giving an idea of their distribution along the *x*-axis. The influence of vaccination status or age at vaccination was not statistically significant. The curve, therefore, represents the marginal effect across all animal categories.

**Figure 5 animals-13-01569-f005:**
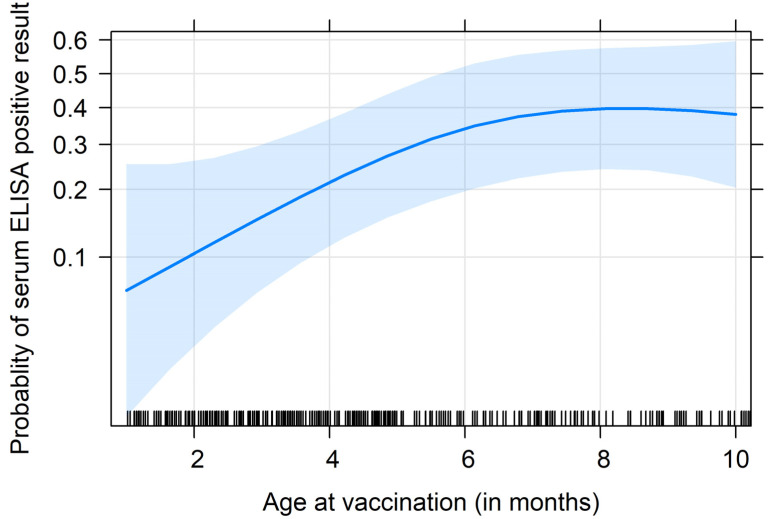
Estimated effect of age at vaccination on the probability of testing positive for Map antibodies at the time of first sampling. The point estimate is represented by the solid blue line, and its 95% confidence interval is shown by the blue area surrounding the line. The black marks on the bottom represent the observations, giving an idea of their distribution along the *x*-axis.

**Table 1 animals-13-01569-t001:** Herds and sampled cows.

Herd	Herd Size ^1^	Control Plan ^2^	Vaccination Plan ^3^	Serological Incidence (%) ^4^	Number of Sampled Cows	Vaccinated Cows	Non-Vaccinated Cows
A	302	February 2003	August 2015	5.7	138	64	74
B	140	February 2009	April 2016	2.5	82	28	54
C	187	February 2007	April 2015	1.8	71	51	20
D	218	September 2006	June 2015	2.2	95	45	50
E	81	October 2012	April 2016	9.1	43	22	21
F	256	August 2002	May 2015	2.1	146	116	30
G	104	November 2000	June 2015	2.1	48	32	16
Overall	1288				623	358	265

^1^ Average number of cattle > 24 months old over the study period. ^2^ Month and year of entrance in the paratuberculosis control plan managed by the GDS de la Meuse. ^3^ Month and year of entrance in the paratuberculosis vaccination plan managed by the GDS de la Meuse. ^4^ Serological incidence in the year of vaccination initiation in ≥ 24-month-old cows.

**Table 2 animals-13-01569-t002:** Distribution of vaccinated and non-vaccinated cows by sampling frequency.

	Number of Samples during the Study Period				
	1 (% ^1^)	2 (%)	3 (%)	4 (%)	Overall
Vaccinated cows	112 (31.3)	212 (59.2)	34 (9.5)	0 (0.0)	358
Non-vaccinated cows	60 (22.6)	153 (57.7)	49 (18.5)	3 (1.1)	265
Overall	172 (27.6)	365 (58.6)	83 (13.3)	3 (0.5)	623

^1^ Percent are computed row-wise.

**Table 3 animals-13-01569-t003:** Number (%) of fecal samples and cows according to the estimated Map concentration in the feces (in equivalent number of Map per gram of feces).

	Estimated Map Concentration (Equivalent Number of Map·g^−1^)						Total
	Negative	[5–10^2^]	[10^2^–10^3^]	[10^3^–10^4^]	[10^4^–10^5^]	>10^5^	
Vaccinated cows							
Number of samples	464 (72.7)	130 (20.4)	20 (3.1)	15 (2.4)	6 (0.9)	3 (0.5)	638
Number of cows ^1^	221 (61.7)	97 (27.1)	18 (5.0)	13 (3.6)	6 (1.7)	3 (0.8)	358
Non-vaccinated cows							
Number of samples	420 (80.5)	74 (14.2)	16 (3.1)	4 (0.8)	4 (0.8)	4 (0.8)	522
Number of cows ^1^	186 (70.2)	53 (20.0)	14 (5.3)	4 (1.5)	4 (1.5)	4 (1.5)	265
Overall							
Number of samples	884 (76.2)	204 (17.6)	36 (3.1)	19 (1.6)	10 (0.9)	7 (0.6)	1160
Number of cows ^1^	407 (65.3)	150 (24.1)	32 (5.1)	17 (2.7)	10 (1.6)	7 (1.1)	623

^1^ For a given cow, the highest estimated concentration found across its fecal samples was used.

## Data Availability

The data presented in this study are provided as Appendix A.

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
