# Peer review of "Effects of Silirum®-Based Vaccination Programs on Map Fecal Shedding and Serological Response in Seven French Dairy Herds"

_animals, 2023, doi:10.3390/ani13091569_

Round 1
Reviewer 1 Report
congratulations - well articulated, novel research, appropriate design and conclusions. Please see file for suggestions on rewording/typos

Author Response
Response to reviewer 1
We thank reviewer 1 for his helpful comments and suggestions that improved the manuscript.
All suggestions were carefully checked and considered.
Reviewer 2 Report
THIS STUDY SHOWS THE RESULTS OF A FRENCH Map CONTROL PLAN USING VACCINATION. IMPLEMENTED IN FIELD CONDITIONS , TIME CONSUMING AND LONG LASTING STUDY , THE MAJOR DIFFICULTY WAS THE BIAS CONTROL, CLEARLY PERFORMED HERE .
HOWEVER SOME DETAILS HAVE TO BE MODIFIED :
INTERNATIONAL UNITS HAVE TO BE USED : gr ARE NOT g AND 10-1 IS NOT 10-1
L 223 strain ATCC 19698 from 108 to 101 copies
estimated Map concentration ≥ 1000.gr-1 of faeces) (Table 3, Figure 1)
L 237 Map per gram of faeces (log10 Map.gr-1).
L 329 (i.e. 8 Map.gr-1 of faeces)
L 339 (in log10 equivalent Map.gr-1)
L 383 estimated concentration of Map > 100 gr-1
for 30 seconds at 6800 rpm three times in a bead beater (IE IN g ?)
L 338 Fig I : Clin. cow) and a vaccinated clinically healthy but persistent shedding ewe (Sub. ewe).
L 369 Fig 1a
L376 Fig 1b
Figure 4. Estimated effect of age at sampling on the amount of Map shed in the faeces. ( IT WOULD BE USEFULL TO INDICATE IN VACCINATED OR NOT AS AGE IS A CONFUSING FACTOR)
2.14.0.0 2.14.0.0
Author Response
Response to reviewer 2
We thank reviewer 2 for his helpful comments and suggestions that improved the manuscript.
All suggestions were carefully checked and considered.
R : L 223 strain ATCC 19698 from 108 to 101 copies
estimated Map concentration ≥ 1000.gr-1 of faeces) (Table 3, Figure 1)
L 237 Map per gram of faeces (log10 Map.gr-1).
L 329 (i.e. 8 Map.gr-1 of faeces)
L 339 (in log10 equivalent Map.gr-1)
L 383 estimated concentration of Map > 100 gr-1
for 30 seconds at 6800 rpm three times in a bead beater (IE IN g ?)
L 338 Fig I : Clin. cow) and a vaccinated clinically healthy but persistent shedding ewe (Sub. ewe).
L 369 Fig 1a
L376 Fig 1b
A: Standard units and references to figures and tables have been checked and corrected where necessary.
R: “Figure 4. Estimated effect of age at sampling on the amount of Map shed in the faeces. (IT WOULD BE USEFULL TO INDICATE IN VACCINATED OR NOT AS AGE IS A CONFUSING FACTOR)”
A: For clarity, the following sentence has been added to the Figure 4 caption: “The influence of vaccination status or age at vaccination was not statistically significant. The curve, therefore, represents the marginal effect across all animal categories.”
Reviewer 3 Report
Dear Authors - the review is attached.

Author Response
Response to reviewer 3
We thank reviewer 3 for his helpful comments and suggestions that improved the manuscript.
All suggestions were carefully checked and considered.
R: Lines 14 and 30: Consider simplifying this sentence by removing “odd” and re-writing this to read “A statistically significant reduction of faecal shedding was . . . . “
A: these sentences have been modified.
R: Line 20: The “p” in paratuberculosis should not be capitalized.
A: corrected
R: Lines 33-34: For clarity, replace the word "youngest" with the word "younger"
A: corrected
R: Line 41: delete one of the words "Paratuberculosis"
A: corrected
R: Line 43: For clarity, replace the word "ruminant" with the word "ruminants"
A: corrected
R: Line 59-62: This sentence needs clarifying. Are the authors trying to say “However, successful disease control requires the commitment of owner and herd manager over many years. When effective changes are lacking or reduced, clinical cases of PTB continue to occur years and even . . . .”
A: thank you for this suggestion. The sentence has been corrected.
R: Line 95: replace the word "age" with the word "ages"
A: corrected
Line 117: For clarity, replace the word "Herd" to the word "Herds"
A: corrected
Line 117-118: the infectious agent is MAP. Please replace the word “paratuberculosis” with the
abbreviation “MAP”.
A: corrected
Line 127: For clarity, replace the word "adult" with the word "adults"
A: corrected
Line 130: For clarity, replace the word "occasion" with the word "occasions"
A: corrected
Line 147: replace the word “flock” with the word “herd”
A: corrected
Line 310: For clarity, please delete one of the "was" words
A: corrected
Line 313: For clarity, replace the word "differed" with the word "differ"
A: corrected
Line 352: For clarity, replace the word "herd" to the word "herds"
A: corrected
Line 360: For clarity, include the word "ratio" to read “on the odd ratio of MAP shedding…” “
Please make this change though out the text where “odd” or “odds” is mentioned.
A: corrected when needed.
Line 420: delete the word “cows” to read “The proportion of vaccinated cows that . . . .”
A: corrected
Line 447: replace the word “heard” with the word “herd”.
A: corrected
Line 480: replace the word “biases” with the word “bias”
A: corrected
R: Supplementary: Why are the columns, AGE AT VACCINATION (MONTHS), DAYS IN MILK (MONTHS), AGE AT SAMPLING (MONTHS) and ELISA S/P VALUE have values to 8 or 9 decimal places? Surely the measurements are not that sensitive? Please amend these data sets to mirror the sensitivity of the measurement read.
A: corrected. The precision has been reduced to the integer or to 1 or 2 decimals if necessary
Reviewer 4 Report
Corbiere and collaborators present an interesting field study on the effects of vaccination with Silirum® in dairy herds in France. Authors evaluate the effect of vaccination on Map fecal shedding and the production of antibodies to Map, concluding that the vaccine maybe beneficial at decreasing fecal shedding when administered at 4-5 months of age.
The manuscript is well written and the strength compared to other similar previosuly published studies is the inclusion of in-herd non-vaccinated controls. The drawback is that each farm enrolls in the vaccination plan in different conditions and then all the data is analzyed together. Some data presented in the supplementary table can be summarized/grouped. A summary table showing the age of sampling after vaccination and/or grouping for the age of vaccination with the results plus and showing Map fecal sheeding of Table 3 divided among vaccinated and non-vaccinated animals would be clearer and help the reader get a better picture of what is happening.
Major concerns
Authors state in lines 106-108 that at the time of the manuscript writing “17 herds have been included in the vaccination program but results on the potential benefit of vaccination are lacking”. I don’t find this statement is necessary specially when only 7 herds have been included in the study. Do authors mean that no benefits have been oboserved in all 17 herds, in the 7 herds that have been included in the study?
Going through the manuscript I’ve had the impression that information was missing. I must say that afterwards in the discussion many of my questions were solved. However, others remain and I feel that the manuscript can be improved if these questions are answered and the information is added and further explained in the manuscript.
I agree that the strength of the study is to have in-herd non-vaccinated controls. However, given the differente scenarios in each farm, maybe data should be analyzed separately for each herd and only herds with close to similar characteristics be grouped. I think that the main difference among farms is the time that has passed between the initiation of the Control Plan and the initiation of the Vaccination Plan. This means that all farms do not have the same starting conditions. It can be assumed that the farms with a long time in the Control Plan should have better starting conditions and in consequence better results after vaccination (less Map elimination), although the vaccination effect in this case might not be as evident as it can be in a farm with a shorter period in the control plan before enrolling in the vaccination plan.
Considering the great difference among herd characteristics, why wasn’t the data from each farm analyzed independently? If this wasn’t done, can it be done now? I believe that real reccomendations for farmers can be formulated for each analyzed farm scenario.
Can data be stratified to show exactly how many months after vaccination was the animal sampled? How were the animals selected for sampling or taking part in the study?
In the M&M section authors state that faeces and blood were collected on 5 occasions between May 2018-Jan 2021 with 6 months intervals (lines 163-164). However, when results are presented authors state that most farms are only sampled once or twice (lines 267-269) and Table 2. How can this be?
Table 3. Why is the data not separated between vaccinated and non-vaccinated animals? or even between farms? This would be helpful.
Regarding serological results, 67,9% remained negative among the vaccinated animals. This seems to be similar to findings in other studies that claim that aprox 30% of vaccinated animals develop a detectable humoral response. Some vaccinated animals (7) seroconvert later on (lines 300-301). It would be nice to know how many months after vaccination these 7 animals become possitive?
Regarding Map shedding, herds C and D (12 and 15 years in control plan before enrolling in vaccination plan, respectively) were associated with lower loads of Map compared to herds F and G (13 and 6 years in control plan before enrolling in vaccination plan, respectively) that were associated with higher loads of Map. I find all this logical except for what happens it herd F. What’s going on in this herd? Could this be attributed to huge management practice differences? I wouldn’t expect this as it has been in a control plan for many years and a lot of the animals in the herd have been vaccinated (near 50%). Is the inclusion of this herd affecting the analysis of all the farms negatively?
Lines 565-567. Humoral response has not really been established as a correlate of protection in PTB so I’m not sure that this statement is correct. The observation that the younger the animals get vaccinated the less the test positive on serum ELISA is OK and can be due to an immature immune system but it doesn’t mean the immune response developed by PTB vaccine in young animals is lower. I would just say it is different. Mostly because you are not considering the effects of the vaccine on the cellular immune response which have been shown to be induced in studies developed by other groups which follow for your convenience:
-Arteche-Villasol N, et al. Early response of monocyte-derived macrophages from vaccinated and non-vaccinated goats against in vitro infection with Mycobacterium avium subsp. paratuberculosis. Vet Res. 2021 May 12;52(1):69.
-Gupta SK et al Mycobacterium avium subsp. paratuberculosis antigens induce cellular immune responses in cattle without causing reactivity to tuberculin in the tuberculosis skin test. Front Immunol. 2023 Jan 18;13:1087015.
Minor concerns
Be careful with spacing, some spaces are missing throughout the document.
Line 41. Paratuberculosis is repeated.
Line 61. Substitute “herds were” with “herds where”.
Line 120. Replace “first to joined” with “first to join”.
Lines 128. Replace “all over 24 months of age cows” with “all cows over 24 months of age”
Line 135. Replace “over than 24 months old” with “over 24 months of age”.
Line 216. Replace “was used of final” with “was used as the final”.
Table 1. Do authors have the month on which the farms started in the Control Plan. If so, this should be added.
Line 313. Replace “did not significantly differed” with “did not significantly differ”.
Author Response
Response to reviewer 4
We thank reviewer 4 for his helpful comments and suggestions that improved the manuscript.
All suggestions were carefully checked and considered.
R: Authors state in lines 106-108 that at the time of the manuscript writing “17 herds have been included in the vaccination program but results on the potential benefit of vaccination are lacking”. I don’t find this statement is necessary specially when only 7 herds have been included in the study. Do authors mean that no benefits have been observed in all 17 herds, in the 7 herds that have been included in the study?”
A: The other 10 herds were not investigated in the present study because they began vaccinating in 2017 or later. The sentence was deleted.
R: Considering the great difference among herd characteristics, why wasn’t the data from each farm analyzed independently? If this wasn’t done, can it be done now? I believe that real recommendations for farmers can be formulated for each analyzed farm scenario.
A: We agree that recommendations for farmers have to be formulated based on each farm’s results. However, analyzing the data separately for each farm would have reduced statistical power and would have reduced our ability to show the effect of vaccination. A herd-specific analysis would have required much larger numbers, which was not possible given the small herd sizes. Our goals were not as specific as that, but to have an overall assessment of the effects of vaccination across a sample of herds.
R: Can data be stratified to show exactly how many months after vaccination was the animal sampled?
A: A supplementary table (Table S1) is now provided with summary statistics of age/time since vaccination at the different sampling points.
R: How were the animals selected for sampling or taking part in the study?
A: For clarity, we modified the M&M section as follows: “Within each herd, a sample of cows was selected for the present study. Selected cows had to be born on the farm and be in their second lactation in the first year of the study (i.e., between May 2018 and 2019).”
R: In the M&M section authors state that faeces and blood were collected on 5 occasions between May 2018-Jan 2021 with 6 months intervals (lines 163-164). However, when results are presented authors state that most farms are only sampled once or twice (lines 267-269) and Table 2. How can this be?
A: We agree that the way the sampling schedule was presented can be confusing. For clarity, the M&M section was modified as follows: “Farms were visited on 5 occasions between May 2018 and January 2021 at 6-month intervals, except the last sampling date which was delayed due to the Covid-19 lockdown in France. However, because not all cows had entered their second lactation by the first two sampling dates (i.e. May and October 2018) and considering the high renewal rate in dairy herds, the total number of samples collected per cow was typically between 1 and 3 (see results section).
R: Table 3. Why is the data not separated between vaccinated and non-vaccinated animals? or even between farms? This would be helpful.
A: Table 3 has been modified so that data are separated between vaccinated and non-vaccinated animals. We also provide detailed information at the farm level in Supplementary Tables S2 and S3.
R: Regarding serological results, 67,9% remained negative among the vaccinated animals. This seems to be similar to findings in other studies that claim that approx 30% of vaccinated animals develop a detectable humoral response. Some vaccinated animals (7) seroconvert later on (lines 300-301). It would be nice to know how many months after vaccination these 7 animals become positive?
A: Actually, among the 246 vaccinated cows that were tested at least once, 19 seroconverted during the study period (lines….). For these cows, Information regarding the time to seroconversion since vaccination has been added as follows: “For these 19 cows, seroconversion was detected on average 38.6 +/- 4.3 months after vaccination (minimum: 32.8 months, maximum: 48.0 months).” The 7 cows you pointed out were already positive at their first sampling date, but their S/P value still increased by more than 50% during the study period.
A: Regarding Map shedding, herds C and D (12 and 15 years in control plan before enrolling in vaccination plan, respectively) were associated with lower loads of Map compared to herds F and G (13 and 6 years in control plan before enrolling in vaccination plan, respectively) that were associated with higher loads of Map. I find all this logical except for what happens it herd F. What’s going on in this herd? Could this be attributed to huge management practice differences? I wouldn’t expect this as it has been in a control plan for many years and a lot of the animals in the herd have been vaccinated (near 50%). Is the inclusion of this herd affecting the analysis of all the farms negatively?
A: We thank the author for these questions as they enabled us to detect errors in the reporting of control plan dates (Table 1), which had been inverted between several herds. Herds C and D spent about 8 and 9 years, respectively, in the Control plan before enrolling in the Vaccination plan. For herds F and G, this duration was about 13 and 15 years, respectively. We do not have any formal explanation as to why herds F and G were associated with higher loads of Map, but this does not appear to be related to time in the Control plan before the implementation of Map vaccination. As mentioned in the discussion section “departures from the best practices may have happened occasionally or repeatedly, which may explain different trajectories at the herd level”. We also added the following sentence: “Other factors that may influence animal immunity, such as nutritional factors, worm burdens, lameness, or the lack of general cleanliness which favors the maintenance and transmission of Map within the herd, could also contribute to differences between farms, but were not investigated.”
A leave one out analysis was performed to evaluate the influence of each herd on model results. Excluding herd For G from the analysis only marginally influenced the estimated effect of the protective effect of vaccination, because in these 2 herds the infection rate is higher in both vaccinated and non-vaccinated cows than in other herds.
R: Lines 565-567. Humoral response has not really been established as a correlate of protection in PTB so I’m not sure that this statement is correct. The observation that the younger the animals get vaccinated the less the test positive on serum ELISA is OK and can be due to an immature immune system but it doesn’t mean the immune response developed by PTB vaccine in young animals is lower. I would just say it is different. Mostly because you are not considering the effects of the vaccine on the cellular immune response which have been shown to be induced in studies developed by other groups which follow for your convenience:
A: We fully agree that the protective response against intracellular mycobacterial pathogens, such as Mycobacterium avium subsp. paratuberculosis mostly relies on the cell-mediated immune response. The way the antibody response may modulate this protection is poorly established and not well understood. We modified the text as follows “This supports the hypothesis that the antibody response induced by paratuberculosis vaccination in young animals is lower than in older ones due to incomplete development of their immune system. Vaccination also induces a Th1/IL-17 immune response, which is known to play a major role in the protective response against intracellular pathogenic mycobacteria. Our findings, therefore, do not imply that the protective effect of vaccination in young animals is lower, as also supported by the significant protective effect of vaccination on the odds of Map shedding found in cows vaccinated when younger than 4 to 5 months of age.”
Minor concerns
Be careful with spacing, some spaces are missing throughout the document.
Line 41. Paratuberculosis is repeated.
A: Corrected
Line 61. Substitute “herds were” with “herds where”.
A: Corrected
Line 120. Replace “first to joined” with “first to join”.
A: Corrected
Lines 128. Replace “all over 24 months of age cows” with “all cows over 24 months of age”
A: Corrected
Line 135. Replace “over than 24 months old” with “over 24 months of age”.
A: Corrected
Line 216. Replace “was used of final” with “was used as the final”.
A: Corrected
Table 1. Do authors have the month on which the farms started in the Control Plan. If so, this should be added.
A: Table 1 has been updated accordingly and corrected. We thank the reviewer because this update allowed us to detect errors in the reporting of control plan dates, which had been reversed between several herds.
Line 313. Replace “did not significantly differed” with “did not significantly differ”.
A: Corrected